# A Low-Glucose Eating Pattern Improves Biomarkers of Postmenopausal Breast Cancer Risk: An Exploratory Secondary Analysis of a Randomized Feasibility Trial

**DOI:** 10.3390/nu13124508

**Published:** 2021-12-16

**Authors:** Susan M. Schembre, Michelle R. Jospe, Erin D. Giles, Dorothy D. Sears, Yue Liao, Karen M. Basen-Engquist, Cynthia A. Thomson

**Affiliations:** 1Department of Family and Community Medicine, College of Medicine-Tucson, University of Arizona, Tucson, AZ 85721, USA; mjospe@email.arizona.edu; 2Department of Nutrition, Texas A & M University, College Station, TX 77843, USA; egiles@tamu.edu; 3College of Health Solutions, Arizona State University, Tempe, AZ 85287, USA; Dorothy.Sears@asu.edu; 4Department of Kinesiology, College of Nursing and Health Innovation, University of Texas at Arlington, Arlington, TX 76019, USA; Yue.Liao@uta.edu; 5Department of Behavioral Science, University of Texas MD Anderson Cancer Center, Houston, TX 77030, USA; kbasenen@mdanderson.org; 6Department of Health Promotion Sciences, Mel and Enid Zuckerman College of Public Health, University of Arizona, Tucson, AZ 85721, USA; cthomson@arizona.edu

**Keywords:** eating physiology, food intake regulation, blood glucose, metabolism, weight management, obesity, adherence

## Abstract

Postmenopausal breast cancer is the most common obesity-related cancer death among women in the U.S. Insulin resistance, which worsens in the setting of obesity, is associated with higher breast cancer incidence and mortality. Maladaptive eating patterns driving insulin resistance represent a key modifiable risk factor for breast cancer. Emerging evidence suggests that time-restricted feeding paradigms (TRF) improve cancer-related metabolic risk factors; however, more flexible approaches could be more feasible and effective. In this exploratory, secondary analysis, we identified participants following a low-glucose eating pattern (LGEP), defined as consuming energy when glucose levels are at or below average fasting levels, as an alternative to TRF. Results show that following an LGEP regimen for at least 40% of reported eating events improves insulin resistance (HOMA-IR) and other cancer-related serum biomarkers. The magnitude of serum biomarkers changes observed here has previously been shown to favorably modulate benign breast tissue in women with overweight and obesity who are at risk for postmenopausal breast cancer. By comparison, the observed effects of LGEP were similar to results from previously published TRF studies in similar populations. These preliminary findings support further testing of LGEP as an alternative to TRF and a postmenopausal breast cancer prevention strategy. However, results should be interpreted with caution, given the exploratory nature of analyses.

## 1. Introduction

High obesity rates among women in the United States and worldwide are leading to a continued rise in obesity-related cancers, most notably postmenopausal breast cancer [1], which is the leading cause of obesity-related cancer deaths among women in the U.S. [2]. Research shows that excessive weight gain and obesity are significant risk factors for postmenopausal breast cancer among women with and without increased genetic risk [3,4,5,6,7,8,9]. Postmenopausal breast cancer and obesity are linked through insulin resistance—a key modifiable risk factor. By losing weight, women with obesity improve their metabolic- and cancer-related risk biomarkers, including insulin resistance and insulin-signaling adipokines, and circulating pro-inflammatory cytokines that promote tumorigenesis [10,11].

In a seminal Phase II feasibility study of a 6-month intensive lifestyle intervention conducted in postmenopausal overweight and obese women at increased risk for breast cancer, Fabian et al. demonstrated that weight losses of at least 10% effectively reduced serum biomarkers, including insulin resistance (HOMA-IR), at a magnitude that favorably modulated benign breast tissue biomarkers [12,13]. While intensive lifestyle interventions that promote chronic energy restriction, such as that implemented in Fabian’s study, are effective at improving outcomes related to cancer risk in both women at high risk and breast cancer survivors [11,14,15], they are resource-intensive, and similar interventions reported poor long-term adherence. Thus, post-intervention weight regains often hinder long-term treatment effectiveness [16,17,18]. This and other research suggest that alternative weight loss and cancer prevention approaches with clinically meaningful outcomes are essential.

Intermittent fasting paradigms have become increasingly popular among researchers and health-conscious individuals. These eating paradigms aim to align meal-timing with circadian rhythms. Restricting the consumption of energy intake to a daily timespan of 4–10 h (e.g., time-restricted feeding, TRF) enhances synchronization between the central circadian clock (synchronized by light) and peripheral circadian clocks (entrained by nutrient intake) [19,20]. Desynchronization of the central and peripheral circadian clocks was shown to negatively impact insulin sensitivity [21] and beta-cell function [22,23,24]. Compared to chronic energy restriction, human and animal models have shown that TRF reduces metabolic disease risk by improving metabolic homeostasis [25]. Despite published support for TRF to improve metabolic outcomes, meta-analyses of research conducted in women and men (mean age range: 21–77 years) with and without metabolic abnormalities over a median of 6 to 8 weeks concluded that TRF has only modest effects on weight (−1.7 to −0.1 kg) and metabolism [26,27], which could limit its utility as a cancer prevention strategy. Moreover, research and healthcare communities acknowledge that TRF and other fasting paradigms might be inappropriate, unacceptable, or result in lower adherence over time among some individuals [28,29,30]. As such, it is reasonable to explore alternative eating paradigms that are effective and may be more broadly adopted.

Eating when pre-prandial (pre-meal) glucose levels are low (“low-glucose eating pattern”) is an evidence-based strategy to improve maladaptive eating patterns. Research shows that eating without physiological hunger is a modifiable health risk behavior associated with excessive weight gain and increased metabolic risk [31,32]. Consistent with this research, we have shown that individuals with obesity are over-sensitive to changes in glucose levels [32] and that low-glucose eating patterns (defined by personalized thresholds) can be taught as an effective self-regulation strategy that promotes weight control [33,34]. Glucose-guided eating (GGE; historically referred to as hunger training) is a timed eating intervention that teaches people to differentiate between physiological hunger and the hedonic desire to eat [35]. In an intervention setting, individuals taught to eat by the GGE paradigm self-monitor their glucose levels using continuous glucose monitors (CGM) or commercially available glucometers and are instructed to eat when two conditions are met: (a) the desire to eat arises and (b) their glucose levels are at or below their personalized threshold. Typically, this training regimen is implemented for 3–4 weeks while people practicing GGE learn to associate symptoms of hunger with their personalized glucose threshold. The GGE paradigm does not rely on glucose as a valid proxy for hunger for GGE. Rather, it is that, to promote metabolic homeostasis, energy intake should not occur when circulating glucose is the primary source of fuel [32].

The modification of glucose eating patterns by GGE is feasible [33,36] and has resulted in clinically significant, average weight loss of 7.4% in 5 months and improvements in eating behavior (including reductions in hedonic eating) and cancer-related risk biomarkers [34,36,37,38,39]. GGE has resulted in improvements in whole-body insulin sensitivity by 31% (Matsuda index, 7.1 ± 4.1 to 9.4 ± 5.2) in non-diabetic, lean adults (BMI = 23 ± 4 kg/m^2^) [38]. Insulin resistance is the most important modifiable risk factor for postmenopausal breast cancer and is caused by obesity and maladaptive eating patterns. Insulin resistance has downstream effects on insulin signaling (e.g., IGF-1), adipokines (including adiponectin), and circulating pro-inflammatory cytokines that promote tumorigenesis [40,41]. GGE has shown a beneficial effect on insulin sensitivity is greater than that noted in the study by Fabian et al. [12], suggesting that GGE could be more effective at reducing insulin resistance than weight loss alone. Similar to TRF, GGE has an advantage over intensive lifestyle weight loss programs in that it does not promote chronic energy restriction and it requires minimal human resources. This affords GGE the possibility of wide dissemination. There is a great potential benefit of the GGE intervention in postmenopausal breast cancer prevention, and this needs examination.

The key aspect of the GGE intervention is eating when glucose is low, defined as under one’s personalized glucose threshold. The goal of the current study is to explore the impact of low- vs. high-glucose eating patterns on changes in body weight and the selected serum biomarkers of breast cancer risk after 16 weeks and compare these results with those reported in recent TRF studies in similar populations of older women and with the intensive lifestyle intervention conducted by Fabian et al. in postmenopausal overweight and obese women at increased risk for breast cancer. The findings of the current study are intended to support further testing of GGE to promote a low-glucose eating pattern as a strategy to reduce breast cancer risk in postmenopausal women. Therefore, this exploratory, secondary analysis aims to examine the potential effect of a low-glucose eating pattern on postmenopausal breast cancer risk.

## 2. Materials and Methods

Project Take Charge [42] was a 16-week, 2-arm randomized controlled trial in 50 women at risk of postmenopausal breast cancer. Take Charge aimed to assess the feasibility of adding GGE to a highly disseminated, comprehensive weight-loss intervention, the Diabetes Prevention Program (DPP) [43]. As a standalone intervention, the DPP results in weight losses typically observed in traditional weight-loss interventions of 4–7% [44]. In Project Take Charge, it was hypothesized that, if feasible, the addition of GGE to the DPP versus the DPP alone could synergistically improve weight loss and effects on biomarkers of cancer risk similar to earlier work [12]. Forty-six women completed the Take Charge trial (86%), which found that GGE was feasible, but the planned analyses (group × time ANCOVA adjusting for baseline measures) did not result in a synergistic effect when added to the DPP on changes in body weight or the cancer-related serum biomarkers assessed in the parent study, including those reported in the current study [42]. As such, data from women in the DPP-only and DPP + GGE groups were merged. Interestingly, in post-hoc analyses described in the current study, we found that women assigned to both the DPP-only and the DPP + GGE interventions changed their eating patterns in a manner consistent with GGE.

As part of a randomized feasibility study, GGE was added to a 16-week version of the DPP intervention that targeted women at risk of postmenopausal breast cancer (defined, in part, as Gail model lifetime risk > 20% or 5-year risk > 1.66%) [45]. Participants (*N* = 50) were predominantly White, non-Hispanic older women who were well-educated; lived in the Houston, Texas, metropolitan area; and had a BMI > 27 kg/m^2^. This study was approved by the Institutional Review Board and registered at clinicaltrials.gov (NCT03546972). Women provided informed consent prior to initiating the study.

The Project Take Charge protocols were fully described elsewhere [42]. Briefly, as part of Project Take Charge, anthropometric measures (weight and height) and metabolic and cancer risk biomarkers (total cholesterol, HDL, LDL, VLDL, triglycerides, HbA1c, fasting glucose, fasting insulin, insulin resistance by HOMA-IR, CRP, adiponectin, IGF-1) were collected at baseline (week 0) and post-intervention (week 16). Weight (light clothing) and height (without shoes) were measured in duplicate using calibrated equipment to within 0.2 kg and 0.3 cm by trained study staff at baseline, 8 weeks, and 16 weeks. Metabolic and breast cancer risk biomarkers were assessed at baseline and 16 weeks. Fasting blood draws were conducted and processed for analysis according to standardized laboratory protocols at The University of Texas MD Anderson Cancer Center and nearby Labcorp location. Insulin resistance was assessed as HOMA-IR using fasting glucose and insulin levels by the following equation: (Fasting Glucose (mg/dL) X Fasting Insulin (mU/L)/405 [46].

The women enrolled in Project Take Charge additionally provided blinded CGM data using Dexcom G5 (Dexcom, Inc., San Diego, CA, USA) at week 0 (baseline), week 8, and week 16 (post-intervention) for up to 10 days at a time. From the collected CGM data, the mean amplitude of glycemic excursions (MAGE) was calculated using EasyGV [47] as a measure of glycemic variability. The women were trained to record their dietary intake and mealtimes using the combination of a familiar and commercially available diet tracker (MyFitnessPal) and self-captured food photographs shared via email. Diet tracking apps, including MyFitnessPal, were found to be a valid means of assessing energy and nutrient intakes [48,49]. Time-stamped dietary intake was concurrently collected with blinded CGM data for up to 7 days at all three time points. Reported mealtimes were confirmed by the study dietitian using the time-stamped food photos that were matched to MyFitnessPal records. Dietary intake (energy and macronutrient composition) was estimated by transferring the digital diet records into the University of Minnesota Nutrition Data System for Research (NDSR) software. The dietary data transfer was conducted by the study dietitian trained to use NDSR and audited for quality control by the study PI. Dietary records with mealtimes were then merged with the CGM data within 5 min of the time-stamped meals. Discrete eating events were defined as energy intake from foods or beverages of greater than 25 kcals and occurring more than 15 min apart. Women were included in this exploratory analysis if they provided at least 3 valid days of blinded CGM data and time-stamped dietary intake at week 16. A valid day was defined as having at least 2 time-stamped eating events with corresponding CGM data. This resulted in an analytical subgroup of *N* = 19 women.

Women were categorized into eating patterns based on week 16 dietary and CGM data. Those who consumed at least 40% of their recorded meals when their pre-prandial glucose levels were below their personalized threshold will be referred to as following a “low-glucose eating pattern (LGEP)”; whereas those who ate less than 40% of their meals when pre-prandial glucose was below their threshold will be referred to as following a “high-glucose eating pattern (HGEP)”. The threshold of 40% eating events was chosen to define the groups post hoc to maximize between-group differences in reductions of HOMA-IR at 16-weeks. LGEP and HGEP were quantified at all three time points using blinded CGM was calculated as the percentage of reported eating occasions where a participant’s glucose was equal to or less than their computed, personalized threshold (reflected by the average of two, fasted 5 am glucose levels were collected using blinded CGM during the initial week run-in period).

Outcomes between the original intervention groups were similar. Specifically, the intervention groups (DPP-only vs. DPP + GGE) had comparable changes from baseline to 16 weeks in weight (−5.0 kg vs. −4.9 kg) and HOMA-IR (−0.3 vs. −0.4). Therefore, for this analysis, the data from the DPP-only and DPP + GGE groups were combined. The 16-week changes in weight and serum biomarkers were compared between women in the LGEP and HGEP groups using SPSS version 28. The means, standard deviations, medians, and ranges are reported here and compared to findings from Fabian et al. [12] and published TRF studies in comparable study populations [50,51]. The *p*-values or other tests of significance were not reported due to the secondary and exploratory nature of this analysis. TRF studies were identified using MedLine. Only those articles with predominantly older women at or near an age consistent with the onset of menopause (>45 years) were considered.

## 3. Results

Table 1 shows that this analytical sample of women (*N* = 19) were predominantly older, White, non-Hispanic, college-educated women, with a BMI in the obese range at baseline. Out of the 19 women with CGM and dietary data, eight (42%) were identified as following a low-glucose eating pattern at post-intervention (week 16) and categorized in the LGEP group. Women in the LGEP were comparable demographically to those in the HGEP group with modestly greater BMI (Table 1).

At baseline (prior to starting the intervention), nearly 70% of reported eating events occurred when glucose was above fasting levels (“HGEP”). By week 8 in the LGEP group, the majority of reported eating events occurred when glucose levels were below the personalized thresholds (approximately 60%). This change in the LGEP group was maintained at week 16. In the HGEP, fewer reported eating events occurred when glucose levels were below personalized thresholds from baseline to mid-intervention to post-intervention (Figure 1).

Women in the LGEP group experienced notable improvements in adiponectin, HOMA-IR, fasting insulin, and glycemic variability (calculated as the mean amplitude of glycemic excursions or MAGE) (Figure 2). These changes were evident without substantial differences in energy intake (−323 kcal vs. −445 kcal) or weight loss (−7.4 vs. −5.8 kg) for LGEP vs. HGEP at post-intervention week 16 (Appendix A). Additionally, LGEP women showed marked reductions in CGM mean glucose levels, as exemplified in Figure 3.

When results of this study are compared to those from published TRF interventions in similar populations, it suggests that LGEP induces nominally larger average weight loss and reductions in HOMA-IR and fasting insulin that is not explained by changes in energy intake (Table 2). The comparison of LGEP to the intensive lifestyle intervention led by Fabian et al. [12] highlights the potential of an LGEP to induce changes in fasting insulin, HOMA-IR, CRP, and adiponectin, that might similarly translate into favorable modulation of benign breast tissue.

## 4. Discussion

This study supports the potential efficacy of a low-glucose eating pattern (LGEP) to improve metabolic and cancer risk biomarkers, including insulin resistance, in older women. Importantly, these data support a viable alternative to TRF for improving health outcomes. Furthermore, the positive metabolic effects of an LGEP might be achieved without eating all meals under the personalized glucose threshold, further supporting the flexibility of LGEP and the robust effects of LGEP in relation to metabolic health. Specifically, following LGEP at ≥40% of eating events is associated with significant improvements in weight and serum markers of cancer risk over time. These findings are similar to previously reported findings of the GGE intervention, where modest protocol adherence was associated with clinically relevant weight loss and improvements in eating behavior [38]. However, this is the first analysis to examine the association between LGEP and serum biomarkers of breast cancer risk. Importantly, the magnitude of observed improvements in HOMA-IR in response to LGEP was comparable to those previously shown to impact postmenopausal breast cancer risk at the tissue level. We feel these preliminary findings support further testing of LGEP as a breast cancer prevention strategy.

Comparing our results to those from TRF studies suggests that LGEP could be as effective or more effective at reducing the risk of postmenopausal breast cancer. We hypothesize that GGE could effectively teach women to follow LGEP to achieve these outcomes. The results shown here suggest it is worthwhile to conduct a clinical trial aimed at comparing the effects of these interventions on biomarkers of postmenopausal breast cancer. Key features of such a trial should include adherence for a range of population groups and durability of effects after the intervention has ceased. A previous pilot study showed that GGE is acceptable from a patient perspective and outlined adherence barriers and enablers [39]. Further examination and direct comparison of participants’ barriers and challenges to adherence and unwanted side effects in response to GGE and TRF will be needed to confirm GGE as an acceptable alternative to TRF. Comparison of our results to those of Fabian et al. [12] suggests that the magnitude of changes in weight and cancer-related biomarkers produced by LGEP, consistent with GGE, particularly changes in fasting insulin, HOMA-IR, and adiponectin, could have meaningful changes in benign breast tissue indicative of reduced postmenopausal breast cancer risk.

Of note, this and the related Project Take Charge studies to exemplify the benefit of using biological feedback (here glucose levels) to motivate and support effective behavior change (here maladaptive eating patterns). While systematic reviews demonstrated the utility of glucose monitoring in obesity research [52], limited research has been conducted to examine the mechanisms of action by which biological feedback motivates positive health behavior change [53]. One possibility is that GGE may act through the Health Belief Model; wherein, people experience a change in perceived risk by associating their dietary intake to health risk outcomes. Future research will be needed to understand and leverage the use of biological feedback as a cancer prevention strategy better [54].

Strengths of this study include objective quantification of LGEP through passive glucose monitoring and the range of biomarkers tested. However, this analysis is limited by our small, homogeneous sample, which limits the generalizability of our findings. Our findings are most appropriate for hypothesis driving rather than hypothesis testing, and results should be interpreted with caution given the exploratory nature of analyses. Furthermore, the TRF studies were of shorter duration than the current study, which could have implications on comparing the magnitudes of observed effects. It is also unclear why women in Project Take Charge, who were randomized to the DPP-only intervention, changed to their LGEP without additionally receiving the GGE intervention. Future DPP intervention research could test the robustness of these findings. While other clinical trials have tested GGE as a standalone intervention [34,36], following an LGEP, which is promoted by GGE, has not been objectively examined as it was here. As such, it will be important to test the effect of GGE as a standalone intervention (vs. the DPP + GGE) on LGEP and metabolic- and cancer-related biomarkers to ensure the robustness of these preliminary findings in larger and more diverse samples. Furthermore, while 40% of eating events is an achievable change in eating patterns that were sufficient to drive improvements in metabolic outcomes in this group, further research is needed to confirm the adherence level needed for favorable outcomes.

## 5. Conclusions

This exploratory analysis of the impact of LGEP on weight and metabolic markers offers direction for the next steps in testing GGE as an intervention to prevent postmenopausal breast cancer. The adherence goal of 40% offers a feasible target for future GGE interventions and potential for health benefits, most critically a reduction in risk of postmenopausal breast cancer.

## Figures and Tables

**Figure 1 nutrients-13-04508-f001:**
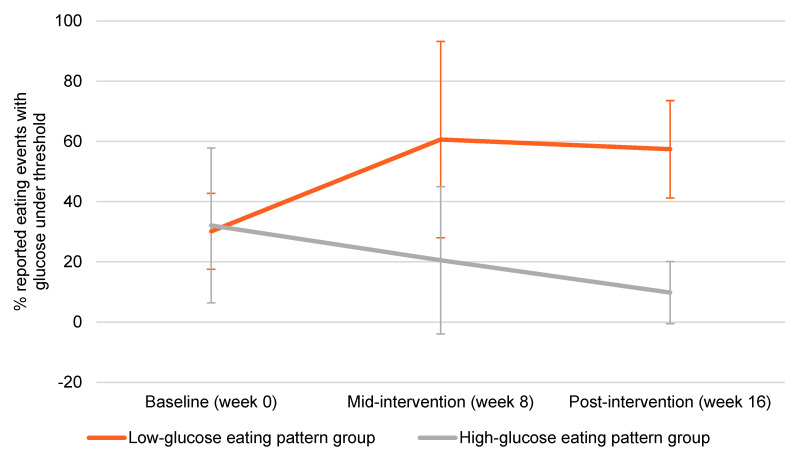
Glucose eating patterns over 16 weeks. Error bars represent standard deviation.

**Figure 2 nutrients-13-04508-f002:**
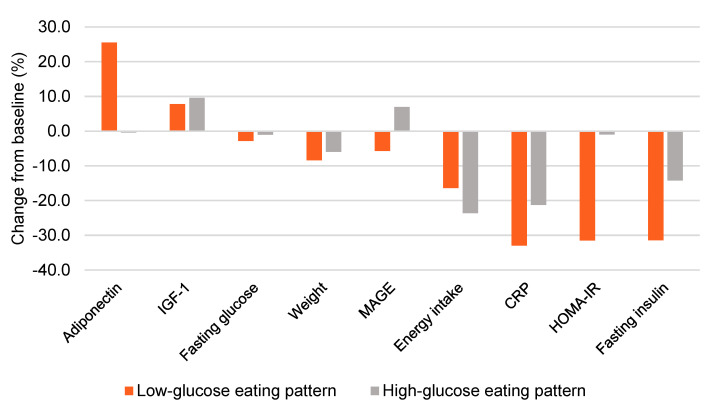
Effect of a low- vs. high-glucose eating pattern on weight, energy intake, and metabolic outcomes at 16 weeks. Bars represented the mean value. IGF-1 = Insulin-like growth factor 1, MAGE = mean amplitude of glycemic excursions, CRP = c-reactive protein, HOMA-IR = homeostasis model assessment-estimated insulin resistance.

**Figure 3 nutrients-13-04508-f003:**
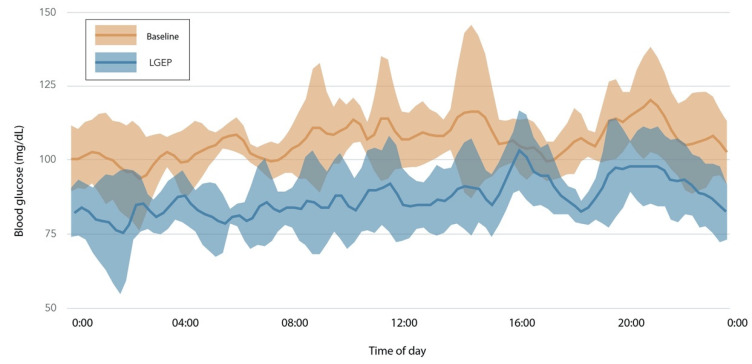
Changes in glucose levels between baseline and 16 weeks for one participant. Summary of 5 days of CGM data at baseline (orange) and 8 days of CGM data at week 8 (blue) after 16 weeks of following a low-glucose eating pattern for a woman with a normal range glycated hemoglobin level (HbA1c = 5.1%) at baseline. Solid lines represent the mean glucose; the shaded areas are standard deviations. This participant followed the low-glucose eating pattern for 18% of eating events at baseline, 91% at week 8, and 76% at week 16. Her baseline fasting glucose of 104 mg/dL and was 84 mg/dL at week 16, a 19% reduction. Her low-glucose eating was present for more eating events than other participants and thus is not representative of all participants, but shows the large change in 2-h glucose levels observed in an individual without prediabetes. low-glucose eating pattern (LGEP).

**Table 1 nutrients-13-04508-t001:** Baseline characteristics of participants according to glucose eating pattern.

	Low-Glucose Eating Pattern	High-Glucose Eating Pattern
*N*	*N* = 8	*N* = 11
DPP + GGE group, *n* (%)	5 (63%)	6 (55%)
White, non-Hispanic, *n* (%)	8 (100%)	11 (100%)
Married, *n* (%)	7 (88%)	11 (100%)
College educated, *n* (%)	8 (100%)	10 (90%)
Age (years)	59.4 ± 7.0	61.9 ± 4.9
Body mass index (kg/m^2^)	32.6 ± 6.2	36.0 ± 7.0

Values are mean ± standard deviation unless otherwise indicated. GGE = Glucose-Guided Eating; DPP = Diabetes Prevention Program.

**Table 2 nutrients-13-04508-t002:** Comparison of LGEP to previously published research in similar populations.

	LGEP (4 Months)	TRF 4HR (2 Months)	TRF 6HR (2 Months)	TRF 8HR (3 Months)	ILI, >10% Weight Loss (6 Months)
	Current study	Cienfuegos, 2020 [50]	Cienfuegos, 2020 [50]	Gabel, 2018 [51]	Fabian, 2013 [12]
*N*	*N* = 7	*N* = 16	*N* = 19	*N* = 23	*N* = 24
Participants	Postmenopausal women at risk for BrCa without DM	90% women	90% women	87% women	Postmenopausal women at risk for BrCa without DM
BMI inclusion (kg/m^2^)	>27	>30	>30	>30	>25
Age (years), mean ± SD	59 ± 7	49 ± 2	46 ± 3	50 ± 2	57 ± 5
Body weight (kg)	−7.4 (−8%)	−3.0 (−3%)	−3.0 (−3%)	−3.0 (−3%)	−12.8, (−16%)
Fasting glucose (mg/dL)	−3.3, (−3%)	−5.0 (−6%)	−2.3 (−2%)	+3 (+4%)	−3.0, (−3.0%)
Fasting insulin (µIU/mL)	−6.6, (−32%)	−2.3 (−19%)	−1.9 (12%)	−2.6 (−31%)	−3.7, (−57%)
Insulin resistance (HOMA-IR)	−0.7, (−32%)	−0.8 (−29%)	−0.5 (−12%)	−0.6 (−38%)	−0.5, (−56%)
IGF-1 (nM)	+7.8, (+8%)	NR	NR	NR	+0.6, (+6%)
Adiponectin	+1.8 (+26%)	NR	NR	NR	+3.5, (+31%)
TNF-α (pg/mL)	NR	−2.4 (−29%)	−0.4 (−3%)	NR	−0.2, (−4%)
CRP (µg/mL)	−0.5 (−33%)	NR	NR	NR	−1.0, (−39%)
Energy intake (kcals)	−323 (−16%)	−528 (−30%)	−566 (−29%)	−341 (−20%)	−387, (−21%)
Macronutrient composition as percentage of energy intake (fat, carbohydrates, protein)	36%, 47%, 18%	36%, 46%, 18%	40%, 40%, 20%	37%, 46%, 17%	20%, 60%, 21%

Values are mean (%) unless otherwise indicated. LGEP = Low-glucose eating pattern; TRF = time-restricted eating; ILI = intensive lifestyle intervention; BrCa = breast cancer; DM = diabetes mellitus; BMI = body mass index; HOMA-IR = homeostasis model assessment-estimated insulin resistance; IGF-1 = Insulin-like growth factor 1; TNF-α = Tumor necrosis factor; CRP = c-reactive protein, NR = not reported.

## Data Availability

The data presented in this study are available on request from the corresponding author.

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
