# Peer review of "A Low-Glucose Eating Pattern Improves Biomarkers of Postmenopausal Breast Cancer Risk: An Exploratory Secondary Analysis of a Randomized Feasibility Trial"

_nutrients, 2021, doi:10.3390/nu13124508_

Round 1
Reviewer 1 Report
This manuscript describes a secondary analysis of a randomized feasibility trial to test a Low-Glucose Eating Pattern among women at risk for postmenopausal breast cancer.
Abstract
Line 24 – There is an extra “Eating”.
Line 26 – There seems to be words missing. ….following a LGEP regimen implemented for >40%
Line 28 – biomarker changes
Line 29 – Suggest rewording – for example, …in overweight or obese women
Line 32 – Suggest rewording TRF as a
Introduction
Line 49 – Suggest rewording as “insulin resistance” is not a biomarker.
Line 55 – Suggests
Lines 62- “causes” language seems too strong here.
Line 67 – Please include the time frames of previous research on weight and metabolic changes for TRF and describe the populations studied.
Line 77 – Consistent with this research
Line 112 & Line 134 – This manuscript (reference 42) is under review; thus, the reader cannot access. That makes it difficult to judge the importance of the current manuscript.
Line 115 – Please provide justification for why the groups were merged. What criteria and/or statistics were used?
Materials and Methods
Line 148 – Dexcom G5?
Line 150 – MAGE?
Lines 155-160 – It would seem that there could be issues with the nutrition data which was input by participants into MyFitnessPal and later transferred into NDSR by a dietitian. Did you evaluate the validity of this approach?
Line 161 – Coffee with cream and sugar could be >25 kcal. How were snacks and beverages handled?
Line 167 – 40% adherence seems a very low bar.
Lines 173-177 – Long and confusing sentence. Please revise.
Lines 178-185 – I’m not convinced that because weight loss and HOMA-IR were similar between these groups that you are justified in combining them.
It’s difficult to tell exactly how many participants were included in the overall feasibility study. Was that also n=19?
I may have missed it, but I don’t see mention of human subjects review and informed consent.
There should be some discussion about the practicality and acceptability among participants for blood using glucose levels in this way. What was the drop out rate in this study?
Author Response
Reviewer #1
This manuscript describes a secondary analysis of a randomized feasibility trial to test a Low Glucose Eating Pattern among women at risk for postmenopausal breast cancer.
Abstract
- Line 24 – There is an extra “Eating”.
This has been removed.
- Line 26 – There seems to be words missing. ….following a LGEP regimen implemented for >40%
We modified the sentence for clarity.
- Line 28 – biomarker changes
We’re unsure what the suggestion is here.
- Line 29 – Suggest rewording – for example, …in overweight or obese women
We have chosen to use person-first language (1), which is recommended by many scholarly journals, government documents, and by the United Nations.
- Line 32 – Suggest rewording TRF as a
We’re unsure what the suggestion is here.
Introduction
- Line 49 – Suggest rewording as “insulin resistance” is not a biomarker.
We changed this to the biomarker that Fabian et al. measured – HOMA-IR, which reflects insulin resistance. This has been modified on line 50.
- Lines 62- “causes” language seems too strong here.
“Causes” has been removed (lines 65-66).
- Line 67 – Please include the time frames of previous research on weight and metabolic changes for TRF and describe the populations studied.
Details on the participants and duration of the studies included in the meta-analyses has been added (lines 64-5).
- Line 77 – Consistent with this research
The typo has been corrected.
- Line 112 & Line 134 – This manuscript (reference 42) is under review; thus, the reader cannot access. That makes it difficult to judge the importance of the current manuscript.
If requested, we can provide a copy of the manuscript which is currently under second review by CAPR.
- Line 115 – Please provide justification for why the groups were merged. What criteria and/or statistics were used?
We have clarified the previously provided justification as follows: “Forty-six women completed the Take Charge trial (86%), which found that GGE was feasible but the planned analyses (group x time ANCOVA adjusting for baseline measures) did not result in a synergistic effect when added to the DPP on changes in body weight or the cancer-related serum biomarkers assessed in the parent study, including those reported in the current study.” (lines 129-33).
Materials and Methods
- Line 148 – Dexcom G5?
Dexcom G5 is the manufacturer and model of the CGM used. We have added the manufacturer name and location for clarity (line 159).
- Line 150 – MAGE?
MAGE is the acronym of mean amplitude of glycemic excursions (MAGE)
- Lines 155-160 – It would seem that there could be issues with the nutrition data which was input by participants into MyFitnessPal and later transferred into NDSR by a dietitian. Did you evaluate the validity of this approach?
While we did not personally evaluate the validity of using MyFitnessPal, previous research has found it valid for measuring dietary intake. To further ensure the reliability of the nutrition data, we opted to enter the MyFitnessPal food records into the NDSR software to improve the accuracy of the energy and nutrient composition of recorded meals. Additionally, the study PI audited the data transfer for quality control. Text has been added (lines 164-5; 172) to clarify this.
- Line 161 – Coffee with cream and sugar could be >25 kcal. How were snacks and beverages handled?
Snacks and beverages with >25 kcal were treated the same as meals, in that they were only allowed under the glucose threshold. We’ve clarified this on lines 174-5
- Line 167 – 40% adherence seems a very low bar.
Yes, while 40% seems low, it reflects an achievable change in eating pattern that was sufficient to drive improvements in metabolic outcomes. However, future research is needed to confirm targeted adherence rates. The limitations and need for further research have been added to lines 308-11.
- Lines 173-177 – Long and confusing sentence. Please revise.
This has been revised for clarity.
- Lines 178-185 – I’m not convinced that because weight loss and HOMA-IR were similar between these groups that you are justified in combining them.
Please see response to point 11.
- It’s difficult to tell exactly how many participants were included in the overall feasibility study. Was that also n=19?
Details about the number of participants recruited and retained in the original Take Charge trial have been added (lines 122, 129-133).
- I may have missed it, but I don’t see mention of human subjects review and informed consent.
The institutional review board statement and informed consent statement were at the end of the document, before the references (lines 324-9).
- There should be some discussion about the practicality and acceptability among participants for blood using glucose levels in this way. What was the dropout rate in this study?
The feasibility of using glucose to guide eating has been shown in multiple studies. A comment about the feasibility and references have been added to line 94. The retention for the Take Charge trial has been added to lines 129. Participants were only included in this secondary analysis if they had CGM and diet measures at week 16. Therefore, it’s not relevant to calculate dropout rate in this subsample since they are all completers.

Reviewer 2 Report
The objective of this paper was to assess the impact of low- vs high-glucose eating on biomarkers of breast cancer risk and to compare these values with previous research. The authors should be commended for composing a well-written and insightful piece of work – it was a pleasure to read. The work is novel and worthwhile, despite the secondary nature of the analysis. My only “major” concerns are regarding clarity of the lack of statistical tests and reporting SD or SEM for all data. The remaining statement are simply suggestions. Specific comments are below; the authors are free to contact me for further clarification.
Introduction: While this is a well written introduction that flows well, it is rather lengthy and detailed. The authors may consider condensing paragraphs 2 & 3 and 4 & 5 – some of this may also be more suited to the discussion. Most of the next-to-last paragraph in the introduction could also go in the methods.
Methods:
Line 172: “The threshold of 40% eating events was chosen to 171 define the groups post-hoc to maximize between-group differences in reductions of HOMA-IR at 16-weeks.” Is this close to the median? Or is there a reference from a previous study further supporting the rationale?
Line 176: Please specify if ‘fasted morning glucose levels’ were the lowest glucose between certain hours or defined by a certain time/event.
Do data not have p-values because of the secondary nature of the analysis? I think this is valid (although others may want p-values from statistical tests), but this should be made clear in the methods.
I may have missed this, but I think an explanation of how the studies in Table 2 were identified would be helpful; even a small statement on a simple search of MedLine or similar would increase the confidence in the results and prepare the reader for Table 2.
Results:
Figure 1 SD bars go off the graph; please correct. Based on the SD bars, the results look significantly different at the end of the intervention, but an asterisk may help a reader easily assess (same for figure 2, if the authors are not with-holding this due to the secondary nature of the analysis)
Figure 2 should have SD or SEM bars
I do think Figure 3 is helpful and appreciate that the authors stated that this is not reflective of all individuals; however, I wonder if there is a way to average this for all participants and perhaps put the individual example as supplementary material.
Discussion:
Overall, there are some excellent points raised and the conclusions reflect the data. Two small points to consider adding: 1) why Gabel et al. 2018 found an increase in fasting glucose and 2) a brief discussion on why/how the particular biomarkers measured relate to breast cancer risk.
Author Response
Reviewer #2
- The objective of this paper was to assess the impact of low- vs high-glucose eating on biomarkers of breast cancer risk and to compare these values with previous research. The authors should be commended for composing a well-written and insightful piece of work – it was a pleasure to read. The work is novel and worthwhile, despite the secondary nature of the analysis. My only “major” concerns are regarding clarity of the lack of statistical tests and reporting SD or SEM for all data. The remaining statement are simply suggestions. Specific comments are below; the authors are free to contact me for further clarification.
Thank you! We have added more detail about statistical tests (lines 129-133; 199-200) and have included SD for all data where possible. We were unable to add SD to Table 2, since it was not provided for the included papers. Instead, we added supplementary table 1 which includes the absolute differences in outcomes from baseline to 16 weeks in low- and high-glucose eating patterns, complete with standard deviations.
- Introduction: While this is a well written introduction that flows well, it is rather lengthy and detailed. The authors may consider condensing paragraphs 2 & 3 and 4 & 5 – some of this may also be more suited to the discussion. Most of the next-to-last paragraph in the introduction could also go in the methods.
We have moved most of the second to last paragraph to the methods section.
Methods:
- Line 172: “The threshold of 40% eating events was chosen to 171 define the groups post-hoc to maximize between-group differences in reductions of HOMA-IR at 16-weeks.” Is this close to the median? Or is there a reference from a previous study further supporting the rationale?
As stated, the 40% threshold was selected based on there being the greatest impact of LGEP on changes in HOMA-IR at 16 weeks in the current sample of women. This threshold reflects a value higher than the sample median of 27.3%. Thanks to this comment we reassessed our decision to categorize the participants at >40% threshold, and discovered a clear break between 27 and 40%. To reassure the reviewers and readers, we recategorized a single participant with 40% adherence as eating according to a low-glucose pattern (and modified the threshold to ≥40% of eating events), which did not appreciably change the results but makes sense so is presented in the paper. We appreciate that this is an exploratory analysis, and have added notes, including in the title, to make this clear to the readers.
- Line 176: Please specify if ‘fasted morning glucose levels’ were the lowest glucose between certain hours or defined by a certain time/event.
5 am CGM levels were used to reflect at least 8 hrs of fasting after 9pm and to ensure the levels were captured before morning intake/breakfast. This has been added to line 190.
- Do data not have p-values because of the secondary nature of the analysis? I think this is valid (although others may want p-values from statistical tests), but this should be made clear in the methods.
Yes, p-values or other tests of significance are not reported due to the secondary and exploratory nature of this analysis. This has been added to lines 199-200.
- I may have missed this, but I think an explanation of how the studies in Table 2 were identified would be helpful; even a small statement on a simple search of MedLine or similar would increase the confidence in the results and prepare the reader for Table 2.
TRF study inclusion was based on a MedLine search, with a focus on study populations for comparison. Only those articles with predominantly older women at or near an age consistent with the onset of menopause (>45 y) were considered. This has been added to lines 201-202.
Results:
- Figure 1 SD bars go off the graph; please correct. Based on the SD bars, the results look significantly different at the end of the intervention, but an asterisk may help a reader easily assess (same for figure 2, if the authors are not with-holding this due to the secondary nature of the analysis).
Figure 1 has been corrected. As noted in item 27, we are not presenting p-values.
- Figure 2 should have SD or SEM bars
We have instead provided a supplementary table 1, for readers who are interested in the absolute differences in outcomes from baseline to 16 weeks in low- and high-glucose eating patterns, complete with standard deviations.
- I do think Figure 3 is helpful and appreciate that the authors stated that this is not reflective of all individuals; however, I wonder if there is a way to average this for all participants and perhaps put the individual example as supplementary material.
This is a great idea, but unfortunately problematic as the large variety in eating times and patterns makes it difficult to see patterns other than the mean shift of glucose, which is better understood through the mean results presented in table 2. The best way to demonstrate the possible changes is at the individual level.
Discussion:
Overall, there are some excellent points raised and the conclusions reflect the data. Two small points to consider adding: 1) why Gabel et al. 2018 found an increase in fasting glucose and 2) a brief discussion on why/how the particular biomarkers measured relate to breast cancer risk.
While Gabel et al. found a small increase in fasting glucose (a change from 79 to 82 mg/dl), this was probably due to chance. The authors did not comment on this in their paper. We added an explanation of the link between insulin resistance, biomarkers, and breast cancer in the introduction, lines 99-103.